# A pH-sensitive switch activates virulence in *Salmonella*

**Dasvit Shetty[1], Linda J Kenney[1,2,3]\***

[1]Mechanobiology Institute, National University of Singapore, Singapore, Singapore; [2]Department of Biochemistry and Molecular Biology, The University of Texas Medical Branch at Galveston, Galveston, United States; [3]Sealy Center for Structural Biology and Molecular Biophysics, The University of Texas Medical Branch at Galveston, Galveston, United States

**Abstract** The transcriptional regulator SsrB acts as a switch between virulent and biofilm life-styles of non-typhoidal *Salmonella enterica* serovar Typhimurium. During infection, phosphorylated SsrB activates genes on *Salmonella* Pathogenicity Island-2 (SPI-2) essential for survival and replication within the macrophage. Low pH inside the vacuole is a key inducer of expression and SsrB activation. Previous studies demonstrated an increase in SsrB protein levels and DNA-binding affinity at low pH; the molecular basis was unknown (Liew et al., 2019). This study elucidates its underlying mechanism and in vivo significance. Employing single-molecule and transcriptional assays, we report that the SsrB DNA-binding domain alone (SsrBc) is insufficient to induce acid pH-sensitivity. Instead, His12, a conserved residue in the receiver domain confers pH sensitivity to SsrB allosterically. Acid-dependent DNA binding was highly cooperative, suggesting a new configuration of SsrB oligomers at SPI-2-dependent promoters. His12 also plays a role in SsrB phosphorylation; substituting His12 reduced phosphorylation at neutral pH and abolished pH-dependent differences. Failure to flip the switch in SsrB renders *Salmonella* avirulent and represents a potential means of controlling virulence.

**\*For correspondence:** likenney@utmb.edu

**Competing interest:** The authors declare that no competing interests exist.

## Editor's evaluation

*Salmonella* invades and survives in host cells via SPI-1 and SPI-2 type 3 secretion system mechanisms, with the SPI-2 system allowing for intracellular survival in *Salmonella*-containing vacuoles, which have a low-pH environment. Transcription of SPI-2 genes at low pH is activated by the DNA-binding SsrB protein, which sits at the top of the SPI-2 regulatory hierarchy. This important study provides convincing insights as to how SsrB is allosterically affected by pH, resulting in acid-dependent DNA binding.

## Introduction

*Salmonella enterica* serovar Typhimurium is a pathogen that causes gastroenteritis in humans and a typhoid-like disease in the mouse. *Salmonella* pathogenicity is largely conferred by the presence of horizontally acquired virulence genes encoded within genomic regions called *Salmonella* pathogenicity islands (SPIs) (*Hensel, 2000*; *Lee et al., 1992*). The most well-characterized genomic islands are SPI-1 and SPI-2, which encode two distinct type 3 secretion systems (T3SS), as well as genes encoding secreted effectors that are important for pathogenesis. The SPI-1 T3SS aids in the initial attachment and invasion of the intestinal epithelium (*Galán and Curtiss, 1989*; *Mills et al., 1995*), while SPI-2 genes play an essential role in the survival of *Salmonella* within the macrophage vacuole and its subsequent maturation into a *Salmonella*-containing vacuole (SCV) (*Cirillo et al., 1998*; *Feng et al., 2003*; *Kuhle and Hensel, 2004*; *Lee et al., 2000*; *Ochman et al., 1996*; *Shea et al., 1996*).

Regulation of the SPI-2 pathogenicity island is complex and involves silencing by the nucleoid-associated protein H-NS (*Gao et al., 2017*; *Liu et al., 2010*; *Lucchini et al., 2006*; *Winardhi et al., 2015*) and anti-silencing by response regulators (RRs) (*Desai et al., 2016*; *Walthers et al., 2011*; *Will et al., 2014*). RRs are part of a signal transduction system prevalent in bacteria. Such two-component systems consist of a membrane-bound histidine kinase and a cytoplasmic RR, which binds to DNA and activates gene transcription (*Hoch and Silhavy, 1995*; *Kenney and Anand, 2020*). The SsrA/B system plays a crucial role in regulating SPI-2 gene expression (*Feng et al., 2003*; *Gao et al., 2017*; *Kenney, 2019*; *Lee et al., 2000*). Activation of SPI-2 genes requires phosphorylation of the SsrB RR on a conserved aspartic acid residue by its kinase SsrA (*Carroll et al., 2009*; *Feng et al., 2004*). Upon activation, SsrB binds to AT-rich regions of DNA and activates transcription of SPI-2 promoters via displacement of the nucleoid-binding protein H-NS (*Walthers et al., 2011*), as well as direct recruitment of RNA polymerase (*Walthers et al., 2007*). The expression of *ssrAB* is surprisingly complex; a promoter for *ssrB* resides in the coding region of *ssrA*, a 30 bp intergenic region lies between *ssrA* and *ssrB*, and both genes have extensive untranslated regions, suggesting post-transcriptional or translational control (*Feng et al., 2003*). Each component of the enigmatic SsrA/B system is regulated by separate global regulators EnvZ/OmpR (*Feng et al., 2003*; *Lee et al., 2000*) and PhoQ/P (*Bijlsma and Groisman, 2005*), indicating an uncoupling of the operon. This complexity was confounding, but recent studies demonstrated a non-canonical role for unphosphorylated SsrB in the absence of its kinase SsrA in driving biofilm formation and establishment of the carrier state, indicating a dual function for SsrB in controlling *Salmonella* lifestyles (*Desai et al., 2016*; *Desai and Kenney, 2017*). Recently, we counted SsrA and SsrB molecules using photoactivation localization microscopy (PALM) and demonstrated their uncoupling and stimulation by acid pH (*Liew et al., 2019*). This complex hierarchy of gene activation ensures that activation of SPI-2 occurs only under conditions that presumably mimic the macrophage vacuole such as low pH, low $Mg^{2+}$, and high osmolality (*Chakraborty et al., 2015*; *Choi and Groisman, 2016*; *Deiwick et al., 1999*; *Miao et al., 2002*).

Upon encountering the acidic environment of the vacuole, *Salmonella* acidifies its cytoplasm in an OmpR-dependent manner through repression of the *cadC/BA* system (*Chakraborty et al., 2015*). Intracellular acidification provides an important signal for expression and secretion of SPI-2 effectors. There is now increasing evidence that this change in intracellular pH is important for pathogenesis (*Chakraborty et al., 2015*; *Chakraborty et al., 2017*; *Choi and Groisman, 2016*; *Kenney, 2019*; *Liew et al., 2019*), although little is known as to how cytoplasmic acidification leads to SPI-2 gene activation. In particular, the effect of acidification on SsrB has not been thoroughly investigated until now.

In this study, we demonstrate that acid-stimulated DNA binding by SsrB is not a property of the C-terminal DNA-binding domain but is allosterically driven by a single conserved histidine residue in the receiver domain. Acid pH drives a highly activated state of SsrB that mimics the high-affinity phosphorylated state of other RRs such as OmpR (*Head et al., 1998*). A mutant substituting histidine 12 (His12) with a glutamine residue retained 100% of the wild-type transcriptional activity at neutral pH and eliminated activity at acid pH. Substitution of His12 with an aromatic amino acid retained pH sensitivity, but substantially reduced its activity compared to the wild-type. In addition to influencing SsrB pH sensing, His12 had minor effects on SsrB phosphorylation. Eliminating acid sensitivity renders *Salmonella* avirulent and thus represents a potential target for controlling infection.

## Results

### Acid pH increases SsrB affinity for DNA

Previous studies used single-particle tracking PALM (spt-PALM) and demonstrated an acid-dependent increase in SsrB binding to DNA in single cells (*Liew et al., 2019*). Hence, we were interested in determining the precise changes of a SPI-2 promoter containing a known SsrB-binding site in response to acid pH. The SPI-2 promoter *sseI* (formerly known as *srfH*) contains an SsrB-binding site of ~47 bp, as determined by DNase I footprinting (*Feng et al., 2004*). We used this 47 bp region to construct a DNA hairpin in order to measure SsrB-binding affinity using a single-molecule unzipping assay (*Gulvady et al., 2018*, see 'Materials and methods'). We compared SsrB binding at neutral pH (7.4) and acid pH (6.1), the pH that we determined was the intracellular pH ($pH_i$) of *Salmonella* in the SCV (*Chakraborty et al., 2015*; *Chakraborty et al., 2017*). At pH 7.4, the binding of SsrB to *sseI* was extremely cooperative (Hill coefficient ($n$) = 8 ± 1) and the dissociation constant ($K_D$) was ~148 ± 2 nM (*Figure 1a*). At

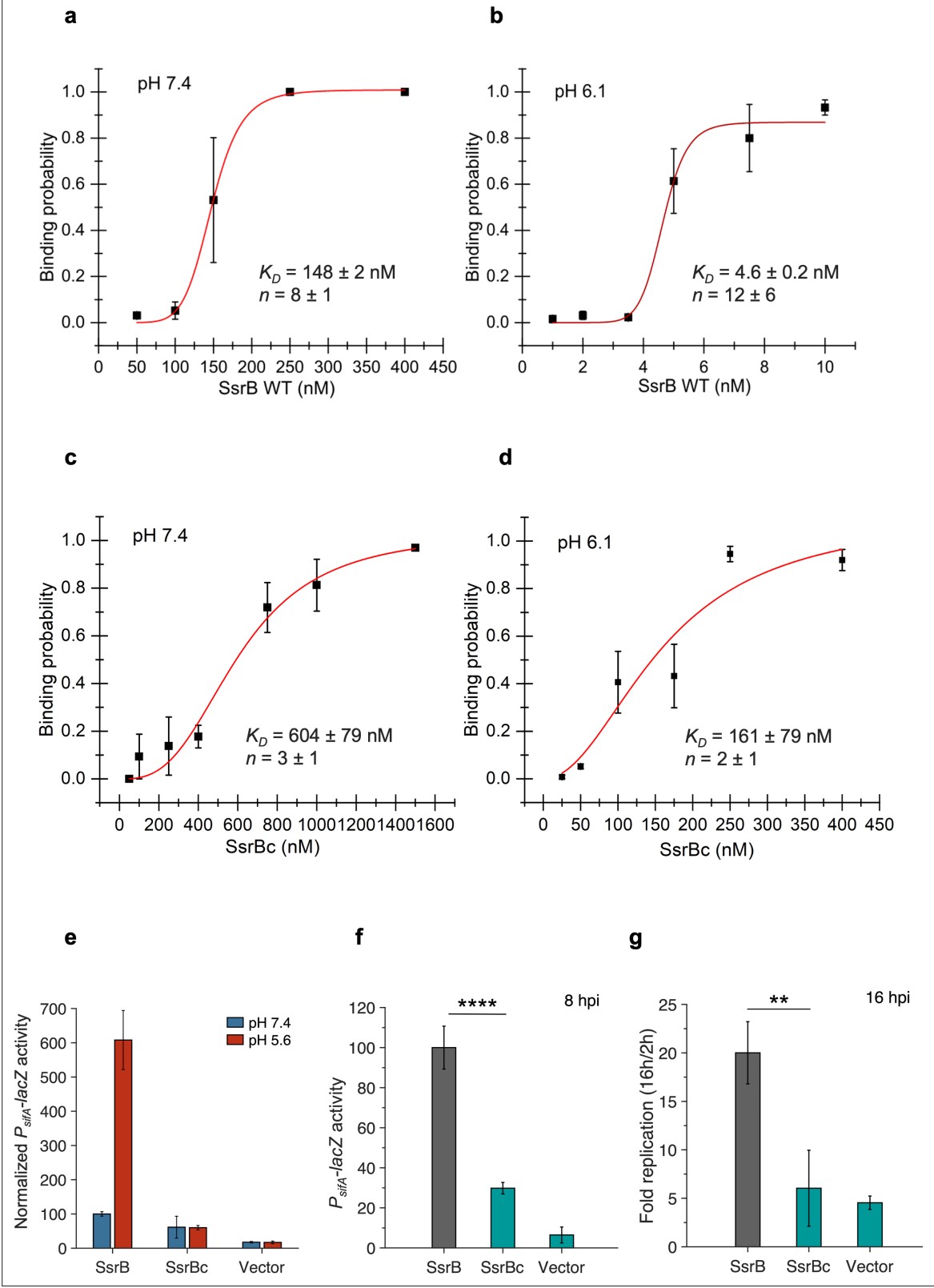

**Figure 1.** SsrBc is not the locus of acid-sensitive DNA binding. (**a**) The plots represent the binding probability of SsrB or SsrBc to the *sseI* DNA hairpin as a function of protein concentration (nM) at pH 7.4 and 6.1. At neutral pH, SsrB binds to the *sseI* promoter with a $K_D$ of 148 ± 2 nM, Hill coefficient (n) = 8 ± 1. (**b**) At acid pH 6.1, the $K_D$ was 4.6 ± 0.2 nM and n = 12 ± 6. (**c**) At neutral pH, SsrBc binds to the *sseI* promoter with a $K_D$ of 604 ± 79 nM, Hill coefficient (n) = 3 ± 1. (**d**) At acid pH 6.1, the $K_D$ was 161 ± 79 nM and n = 2 ± 1. The error bars represent the standard deviation of 3–5 independent

*Figure 1 continued on next page*

Figure 1 continued

measurements. The red line represents the curve derived from fitting the points to the Hill equation to determine the $K_D$. The absence of error bars indicates that the standard deviation was < the symbol. (**e**) $P_{sifA}$-*lacZ* activity in the presence of SsrB or SsrBc grown in magnesium minimal medium (MGM) pH 7.4 (blue bars) and pH 5.6 (red bars) after 3 hr of 0.1% (w/v) arabinose induction. In comparison to SsrB activity at pH 7.4, the activity of the SsrBc strains was reduced to 61 and 60 in pH 7.4 and 5.6 media, respectively. (**f**) $P_{sifA}$-*lacZ* activity measured from strains recovered at 8 hr post infection of HeLa cells. The activity for the SsrBc strain was 30% of the wild-type SsrB strain (***p<0.001, n = 3) (**g**) During HeLa cell infection, the wild-type full-length SsrB strain increased 20-fold after 16 hr post infection (hpi). The SsrBc strain only increased sixfold over the same time period.

The online version of this article includes the following source data and figure supplement(s) for figure 1:

**Source data 1.** Binding probabilities for individual experiments used to create plots for figures 1a-d, $P_{sifA}$-*lacZ* activities and fold replication values for individual experiments used to create plots for *Figure 1e–g*.

**Figure supplement 1.** The plots represent the binding probability of SsrB to the *sseI* DNA hairpin as a function of SsrB concentration (nM) at pH 7.4.

acid pH, the Hill coefficient was relatively unchanged (n = 12 ± 6), while the $K_D$ decreased substantially, to 4.6 ± 0.2 nM (*Figure 1b*). Hence, at the SPI-2 promoter *sseI*, SsrB binding was ~32 times higher affinity at acid pH than at neutral pH and binding at both neutral and acid pH was highly cooperative.

## Acid-stimulated DNA binding does not reside in the DNA-binding domain of SsrB (SsrBc)

SsrB has an N-terminal receiver domain that is phosphorylated and a C-terminal DNA-binding domain, connected by a flexible linker. SsrBc, comprising the linker and DNA-binding domain of SsrB (residues 138–212), has been shown to induce the expression of SPI-2 genes in vitro at a level similar to the full-length SsrB (*Feng et al., 2004*). To determine whether SsrBc alone was responsible for the pH-dependent DNA binding, we used the single-molecule unzipping assay to measure SsrBc binding to the 47 bp *sseI* DNA hairpin at neutral and acid pH. At pH 7.4, the $K_D$ of SsrBc was 604 ± 79 nM, with a Hill coefficient of 3 ± 1 (*Figure 1c*). Thus, the isolated C-terminus of SsrB binds to DNA with a four-fold lower affinity and cooperativity is reduced compared to the full-length protein. At pH 6.1, the $K_D$ value decreased to 161 ± 79 nM, without a substantial change in cooperativity (n = 2 ± 1) (*Figure 1d*). Hence, in comparison to full-length SsrB, SsrBc binds DNA with reduced affinity and cooperativity at both neutral and acid pH. Although the affinity of SsrBc for DNA increased at acid pH, the decrease in $K_D$ was only 4-fold, whereas the full-length SsrB demonstrated a 32-fold decrease over this same pH range. The change in DNA binding at acid pH with SsrBc was similar to the fold change of full-length SsrB at the non-SPI-2 promoter *csgD* (*Liew et al., 2019*).

SsrBc was also incapable of supporting acid-stimulated SPI-2 transcriptional activity compared to full-length SsrB both in vitro and in vivo. Transcriptional activity of the *sifA* promoter (another SPI-2-regulated promoter) was measured in a Δ*ssrB* strain (14028s Δ*ssrB attB::pAH125 P_{sifA}*-*lacZ*, DW637) carrying SsrB or SsrBc on a plasmid under the regulation of an inducible arabinose promoter. Early exponential phase cells grown in SPI-2 non-inducing (pH 7.4) and inducing conditions (pH 5.6) were induced with 0.1% (w/v) arabinose, and the β-galactosidase activity was measured after 3 hr. All activities henceforth described are expressed as the relative $P_{sifA}$-*lacZ* activity of a given SsrB mutant with respect to $P_{sifA}$-*lacZ* activity of the SsrB wild-type-expressing strain grown in magnesium minimal medium (MGM) pH 7.4. For cells grown at pH 7.4, the relative activity of $P_{sifA}$-*lacZ* in the presence of SsrB was set to 100 and the normalized activity of SsrBc was 61. In cells grown at pH 5.6, the activity of the wild-type increased ~6-fold, whereas SsrBc remained at 60 (*Figure 1e*). Thus, while SsrB exhibited a sixfold increase in transcriptional activity when grown in acid pH, transcription by SsrBc was not stimulated at acid pH. Inside HeLa cells, the SsrBc-expressing strain exhibited reduced intracellular survival (16 hr post infection [hpi]) and decreased $P_{sifA}$-*lacZ* activity (8 hpi) compared to SsrB. $P_{sifA}$-*lacZ* activity of the SsrBc strain was 30% and intracellular survival was also 30% compared to the wild-type SsrB strain and similar to the vector-only control (*Figure 1f and g*). Thus, the isolated C-terminal domain was insufficient to support intracellular survival and replication in vivo.

## Analysis and comparison of the NarL/FixJ subfamily of response regulators

Of all of the NarL/FixJ subfamily of RRs, SsrB is the only member for which pH dependence of DNA binding has been reported (*Liew et al., 2019*). SsrB has the highest isoelectric point in this group

**Table 1.** Comparison of some response regulators.

| Protein | Organism | pI | Number of histidines | % identity with SsrB |
|---|---|---|---|---|
| SsrB | *Salmonella* Typhimurium | 7.12 | 9 | 100.0 |
| RcsB | *Escherichia coli* K12 | 6.85 | 3 | 25.5 |
| RcsB | *Salmonella* Typhimurium | 6.85 | 3 | 25.5 |
| EvgA | *Shigella flexineri* | 6.83 | 3 | 28.2 |
| Spo0A | *Bacillus stearothermophilus* | 6.31 | 9 | 25.2 |
| NarL | *E. coli* K12 | 5.73 | 6 | 28.3 |
| DegU | *Bacillus subtilis* | 5.66 | 11 | 26.8 |
| DosR | *Mycobacterium tuberculosis* | 5.62 | 1 | 27.5 |
| VraR | *Staphylococcus aureus* | 5.47 | 5 | 29.2 |
| StyR | *Pseudomonas fluorescens* | 5.42 | 6 | 26.5 |
| LiaR | *Enterococcus faecium* | 5.11 | 5 | 30.0 |
| Spr1814 | *Streptococcus pneumoniae* | 4.95 | 1 | 25.6 |
| PhoP | *E. coli* | 5.10 | 6 | 22.5 |
| OmpR | *E. coli* | 6.04 | 3 | 31.6 |

(pI = 7.12), whereas most of the others range from 4.95 to 6.85 (*Table 1*). An increase in the overall protonation of SsrB as the bacterial cytoplasmic pH decreases within the SCV could contribute to the increase in DNA-binding affinity that we observed (*Figure 1b*). Because SsrBc has an even higher *pI* of ~9.36, it is unlikely that it would undergo enough change in protonation to confer pH sensitivity under SPI-2 inducing conditions, as we also observed (*Figure 1c–e*).

In addition to its high *pI*, SsrB also contains a greater number of histidine residues (9) compared to most other RRs, except DegU (11) and Spo0A (9). Histidine residues are known to play a role in pH sensitivity of numerous proteins (*Furman et al., 2015*; *Mulder et al., 2015*; *Müller et al., 2009*; *Tu et al., 2009*) as the $pK_a$ of the histidine side chain (~6.45) enables it to protonate under the physiological pH range (*Platzer et al., 2014*). The pH threshold of the *Salmonella* cytoplasm below which it expressed and secreted SPI-2 effectors was determined to be between 6.45 and 6.7 (*Chakraborty et al., 2017*; *Kenney, 2019*). Hence, we reasoned that the high number of histidines in SsrB could potentially contribute to its acid-stimulated DNA binding. Since SsrBc alone does not show a very strong pH dependence (*Figure 1c–e*), we screened histidine residues in the N-terminal phosphorylation domain for candidates likely to contribute to the pH sensitivity. We identified four histidine residues in the receiver domain, three of which were unique to SsrB (His28, His34, and His72, *Figure 2—figure supplement 1*). The fourth histidine residue, His12, was relatively well-conserved amongst most of the NarL/FixJ RRs (*Figure 2—figure supplement 1*). Since pH sensitivity has not been explored in these other RRs, we initially screened the three unique histidine residues for pH sensitivity.

## Unique histidines in the receiver domain do not affect pH sensitivity

In an AlphaFold predicted structure of the receiver domain of SsrB (*Jumper et al., 2021*), the three unique histidine residues appear to be solvent exposed, with His28 present in the loop between *α1* and *β2*, His34 is between *β2* and *α2*, and His72 on *α3* (*Figure 2—figure supplement 2*). To assess the effect of these N-terminal histidine residues on SsrB pH sensitivity, each histidine was substituted with alanine using PCR mutagenesis. The single histidine SsrB mutants SsrB H28A, SsrB H34A, and SsrB H72A were assayed for their SPI-2 transcriptional activity when grown in MGM at pH 7.4 or 5.6. There were no substantial differences in the acid-stimulated increase in P*sifA*-*lacZ* activity of any of the singly substituted mutant strains compared to wild-type SsrB (*Figure 2—figure supplement 3a and b*). The increase in P*sifA*-*lacZ* activity at acid pH vs. neutral pH for SsrB H28A, SsrB H34A, and SsrB H72A was 4.2-, 4.8-, and 4.5-fold respectively, similar to the increase of the wild-type (5-fold).

The double histidine mutants SsrB H28A-H34A (SsrB 2H1), SsrB H28A-H72A (SsrB 2H2), and SsrB H34A-H72A (SsrB 2H3) and triple histidine mutant SsrB H28A-H34A-H72A (SsrB 3H) similarly showed no substantial differences in acid stimulation of $P_{sifA}$-lacZ activity in vitro. The increase in $P_{sifA}$-lacZ activity at acid vs. neutral pH for strains expressing SsrB 2H1, SsrB 2H2, SsrB 2H3, and SsrB 3H (4.9-, 6.6-, 5.6-, and 8.1-fold, respectively) was similar to the increase by the wild-type (6-fold) (*Figure 2— figure supplement 3c and d*). In vivo, when the strains expressing SsrB 2H1, SsrB 2H2, or SsrB 2H3 were used to infect HeLa cells, the $P_{sifA}$-lacZ activity at 8 hpi and subsequent survival at 16 hpi were also comparable to the wild-type SsrB. The $P_{sifA}$-lacZ activity measured for SsrB 2H1, 2H2, and 2H3 at 8 hpi was 113, 89, and 94% relative to the wild-type set at 100% (*Figure 2—figure supplement 3e*). Subsequently, intracellular survival at 16 hpi was comparable to the wild-type survival (13.2-fold) (*Figure 2—figure supplement 3f*).

## Conserved histidine 12 is essential for acid-stimulated DNA binding of SsrB

As substitutions of unique histidines failed to abolish the pH sensitivity of SsrB, we screened the remaining N-terminal histidine residue, His12, for pH sensitivity. His12 lies in the loop between β1 and α1 in the receiver domain (*Figure 2a*). It is in close proximity to the phosphorylated residue, Asp56, the metal coordinating residues Asp10 and Asp11, and the polar contact residue Lys106 (*Figure 2a*). We compared the $P_{sifA}$-lacZ activity of alanine and asparagine substitutions at position 12 to wild-type SsrB; they were only 66 and 78% active at pH 7.4 (*Figure 2b*). At pH 5.6, their activity increased by only 1.9- and 1.5-fold, respectively (*Figure 2b and c*). A glutamine substitution at position 12 was 99% active at pH 7.4, while at pH 5.6, there was no increase in activity (0.9-fold). These results indicate that the conserved His12 is the major driver of the acid-stimulated DNA binding of SsrB.

As the glutamine substitution retained full activity at pH 7.4, we used purified SsrB H12Q to measure the effect of His12 substitution on the DNA-binding activity of SsrB in the single-molecule unzipping assay. At pH 7.4, the $K_D$ of binding of SsrB H12Q to the *sseI* promoter was 154 ± 3 nM with a Hill coefficient of 6 ± 1 (*Figure 2d*), similar to wild-type SsrB. At pH 6.1, the $K_D$ remained relatively unchanged at 123 ± 13 nM, while the cooperativity was reduced to 3 ± 1 (*Figure 2e*). This value was similar to the cooperativity of SsrBc (*Figure 1c and d*). Hence, a substitution at H12 eliminated both the increase in DNA-binding affinity and the large change in cooperativity of SsrB at acid pH.

## A role for the aromatic ring in pH sensitivity

To elucidate the mechanism by which His12 contributes to pH sensitivity, additional amino acids were substituted in place of histidine. Histidine can act both as a basic and an aromatic amino acid, hence it can have a variety of interactions, depending on its protonation state (*Liao et al., 2013*). The protonated form of histidine at low pH would theoretically be mimicked by substitution with a positively charged amino acid, while substitution with an aromatic amino acid could mimic the uncharged aromatic imidazole ring. We therefore constructed lysine, tyrosine, and phenylalanine substitutions at His12 and screened the activity and pH sensitivity of these mutants. SsrB H12K $P_{sifA}$-lacZ activity was 34% of wild-type SsrB at pH 7.4 (*Figure 3a*) and its activity only increased 2.7-fold at acid pH (*Figure 3b*). In contrast, both aromatic substitutions exhibited substantial reductions in their activity but maintained pH sensitivity. The activity of H12Y and H12F strains was reduced to 30 and 24%, respectively (*Figure 3a*), but at pH 5.6, these mutants showed a 6.9-fold and 6.7-fold increase in activity, which was comparable to the wild-type (*Figure 3b*). Hence, the lysine substitution at His12 reduced both activity and pH sensitivity in SsrB, while the tyrosine and phenylalanine substitutions led to a reduction in the activity but retained the acid stimulation, indicating a role for the imidazole aromatic ring in the acid stimulation of transcriptional activity.

Substitutions at His12 also led to a reduction in intracellular $P_{sifA}$-lacZ activity and survival of *Salmonella* infections of HeLa cells and RAW macrophages. In HeLa cells, at 8 hpi, the $P_{sifA}$-lacZ activity of SsrB H12Q and H12Y-expressing strains was reduced to 62 and 58% compared to SsrB (*Figure 3c*). Similarly, at 16 hpi, intracellular survival of H12Q and H12Y was reduced to 33 and 40% of the wild-type (*Figure 3d* and *Figure 3—figure supplement 1*). In RAW macrophages, the increase in replication at 16 hpi for *Salmonella*-expressing H12Q was only 2-fold compared to a 13-fold increase observed for the wild-type (*Figure 3—figure supplement 2*). Although the H12Y substitution retained acid pH

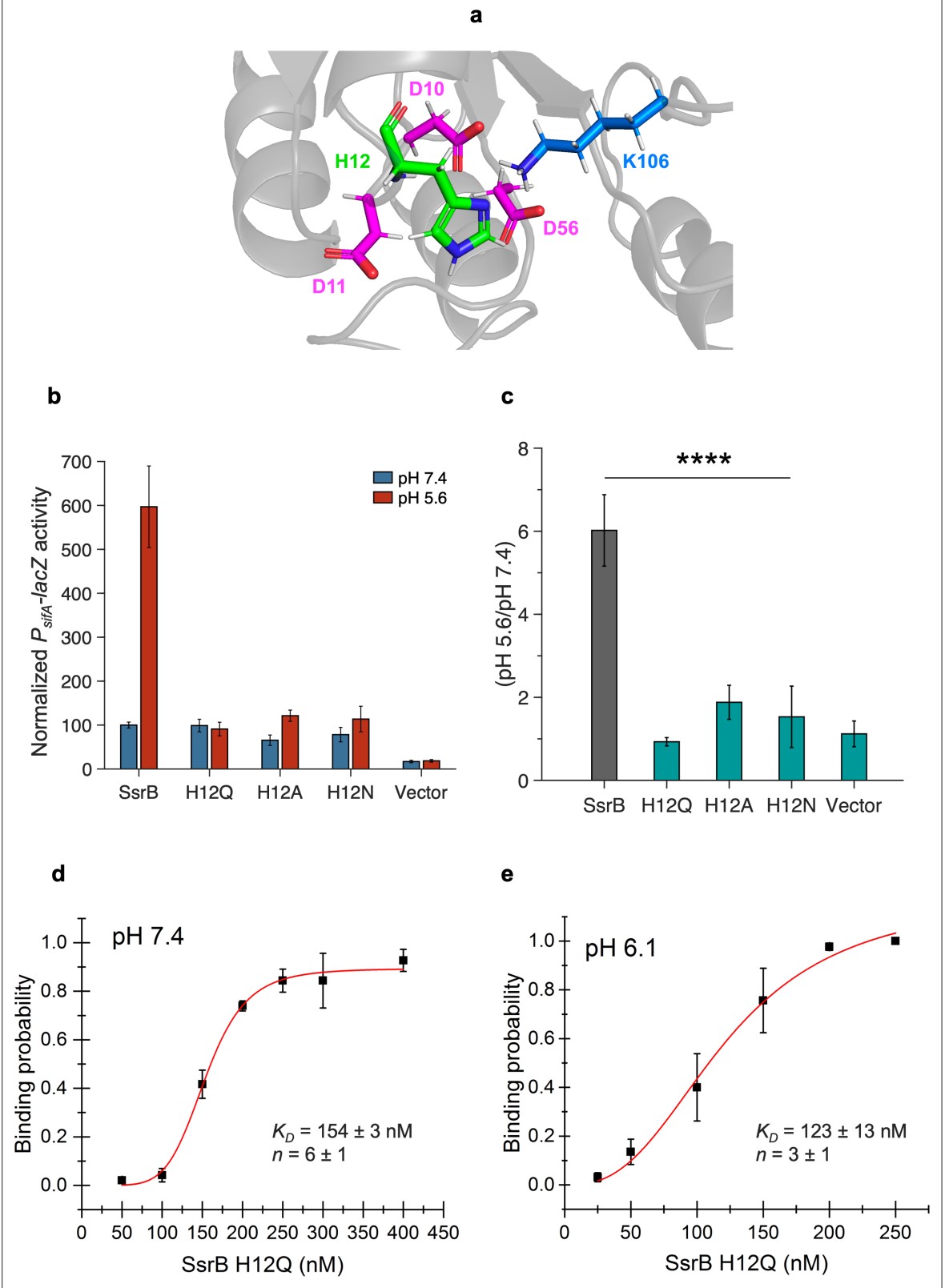

**Figure 2.** Histidine 12 is essential for acid-stimulated DNA binding of SsrB. (**a**) The location of His12, on a predicted receiver domain structure of SsrB. His12 is in the vicinity of the active site residues Asp10, Asp11, Asp56, and Lys106. (**b**) Normalized $P_{sifA}$-*lacZ* activity in the presence of SsrB or H12 mutants of SsrB grown in magnesium minimal medium (MGM) pH 7.4 (blue bars) and pH 5.6 (red bars) after 3 hr of 0.1% (w/v) arabinose induction. (**c**) The H12Q, H12A, and H12N substitutions showed no increase in activity at pH 5.6 (0.9-, 1.9-, and 1.5-fold respectively) compared to wild-type SsrB (6-

*Figure 2 continued on next page*

*Figure 2 continued*

fold) (p<0.0001, n = 3). (**d, e**) The plots represent the binding probability of SsrB H12Q to the *sseI* DNA hairpin as a function of SsrB H12Q concentration (nM) at pH 7.4 and 6.1. (**d**) At neutral pH, SsrB H12Q binds to the *sseI* promoter with a $K_D$ of 154 ± 3 nM, n = 6 ± 1. (**e**) At acidic pH 6.1, the $K_D$ was 123 ± 13 nM and n = 3 ± 1. The error bars represent the standard deviation of 3–5 independent measurements. The red line represents the curve derived from fitting the points to the Hill equation to determine the $K_D$. The absence of error bars indicates that the standard deviation was < the symbol.

The online version of this article includes the following source data and figure supplement(s) for figure 2:

**Source data 1.** $P_{sifA}$-*lacZ* activities for individual experiments used to create plots for *Figure 2b–c*, binding probabilities for individual experiments used to create plots for *Figure 2d–e*.

**Figure supplement 1.** Analysis of NarL/FixJ response regulator (RR) receiver domains and selection of histidine residues.

**Figure supplement 2.** The location of His12, His28, His34, and His72 on the predicted receiver domain of SsrB (visualized using PyMol).

**Figure supplement 3.** Non-conserved histidines in the receiver domain do not confer acid sensitivity to SsrB.

**Figure supplement 3—source data 1.** $P_{sifA}$-*lacZ* activities and fold replication values for individual experiments used to create plots for *Figure 2— figure supplement 3*.

sensitivity in vitro, the low intracellular transcriptional activity and survival can be attributed to the overall reduction in its activity compared to the wild-type.

## The effect of His12 substitution on SsrB phosphorylation

As His12 is proximal to the phosphorylated residue Asp56 (*Figure 2a*), we examined the effect of His12 substitution on SsrB phosphorylation. SsrB is readily phosphorylated in vitro by the small molecule phosphodonor phosphoramidate (PA) (*Feng et al., 2004*), we thus compared the phosphorylation of wild-type SsrB with the H12Q mutant. The amount of phosphorylated protein (SsrB~P or H12Q~P) generated was determined by resolving the reaction mixture on a $C_4$ reverse-phase HPLC column under a 40–50% (v/v) acetonitrile gradient (*Feng et al., 2004*). 12 μM of SsrB or H12Q was phosphorylated for 10 min at varying concentrations of PA at pH 7.4. The wild-type protein was readily phosphorylated, reaching 50% phosphorylation with 2.9 mM PA ($K_{0.5}$) (*Figure 4a*, squares). By comparison, H12Q was phosphorylated more slowly and required higher PA concentration at saturation ($K_{0.5}$ = 7.1 mM PA; *Figure 4a*, triangles). We next compared wild-type and H12Q phosphorylation at varying reaction times with higher PA (15 mM) at neutral and acid pH (*Figure 4b and c*). Wild-type SsrB exhibited a 3.2-fold reduction in the $K_{obs}$ for phosphorylation at acid pH compared to neutral pH ($K_{obs}$ 0.62 and 0.19 min⁻¹, respectively). In contrast, there was no difference in H12Q phosphorylation between neutral and acid pH ($K_{obs}$ 0.20 and 0.17 min⁻¹, respectively). Thus, substitution of His12 both reduced phosphorylation at neutral pH and abolished pH-dependent differences in phosphorylation. An interesting observation was the delayed retention time of H12Q compared to SsrB, irrespective of the pH of the phosphorylation buffer, suggestive of conformational differences between the two proteins (*Figure 4—figure supplement 1*).

## Discussion

While the effector domains of RRs confer target promoter specificity, it is the receiver domains that fine-tune their activation in response to environmental signals. In the case of SsrB, the receiver domain senses a reduction in pH to allosterically increase DNA-binding affinity of the C-terminal effector domain. Acid pH serves as an activation signal that effectively puts SsrB in a high-affinity DNA-binding mode, similar to the role phosphorylation plays in many RRs. SsrBc alone had a fourfold lower DNA-binding affinity than SsrB, and this affinity only increased by fourfold in acidic pH in vitro (*Figure 1c and d*). In vivo, SsrBc resulted in lower SPI-2 induction and no increase in SPI-2 activity in acidic medium. Correspondingly, SsrBc did not support optimal SPI-2 activation and intracellular replication during infection of HeLa cells or macrophages (*Figure 1f and g*). Therefore, although SsrBc was capable of DNA binding, it was insufficient to confer pH sensing or full activity during infection.

Interestingly, while we previously demonstrated a 5-fold acid-stimulated increase in the DNA-binding affinity of SsrB to the ancestral *csgD* promoter (i.e. non-SPI-2; *Liew et al., 2019*), this work revealed a 32-fold increase in its affinity to the SPI-2 promoter *sseI*. This finding was also in agreement with previous AFM studies, which showed abundant binding of SsrB to the SPI-2 promoter *sifA* at acid pH compared to almost no binding observed at neutral pH. Furthermore, the AFM images indicated

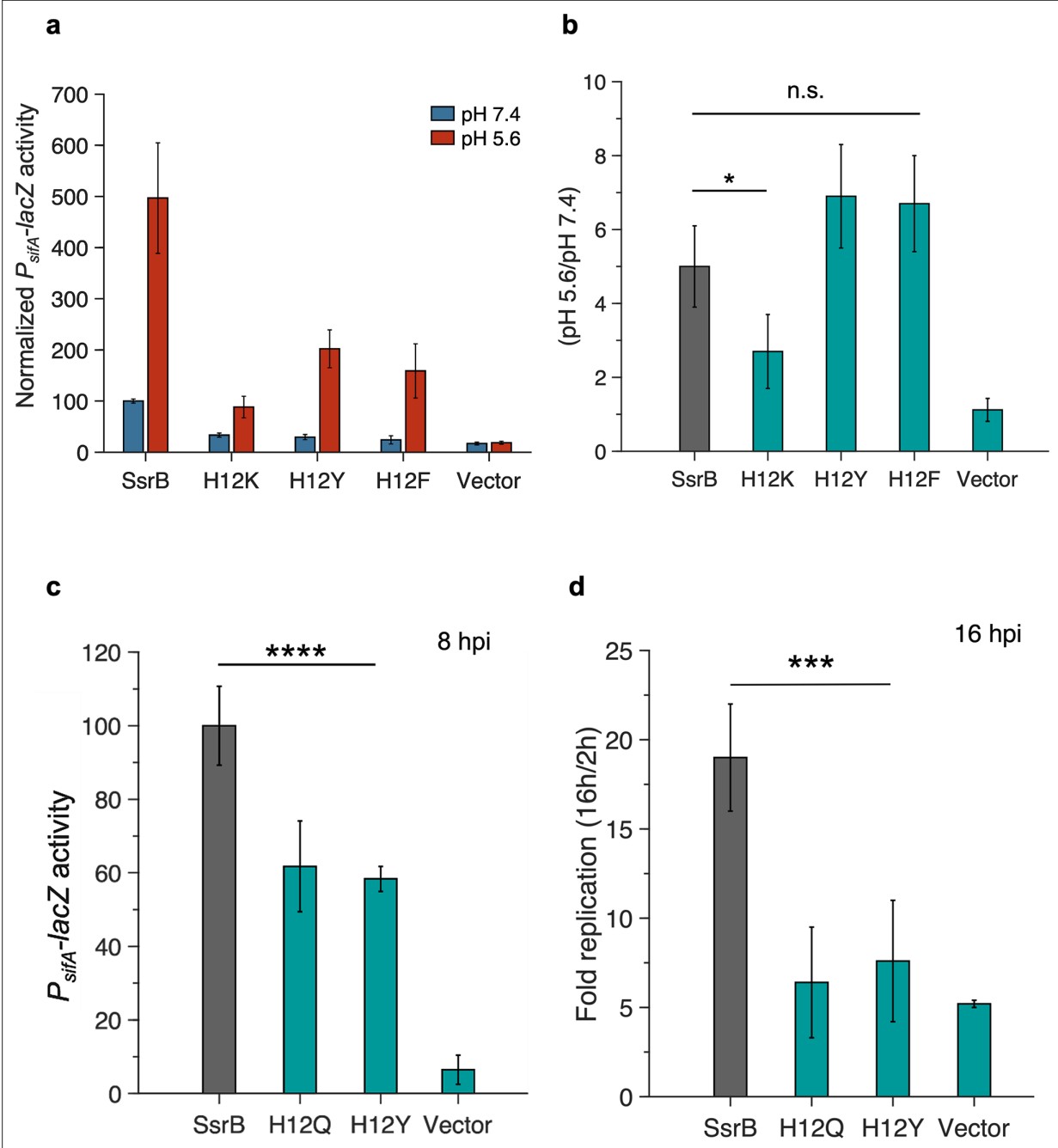

**Figure 3.** Aromatic substitutions at position 12 retain SsrB pH sensitivity. (**a**) Normalized $P_{sifA-lacZ}$ activity in the presence of SsrB or H12 mutants of SsrB grown in magnesium minimal medium (MGM) pH 7.4 (blue bars) and pH 5.6 (red bars) after 3 hr of 0.1% (w/v) arabinose induction. The H12K, H12Y, and H12F substitutions showed a decrease in activity at pH 7.4 compared to SsrB (34 ± 4%, 30 ± 4%, and 24 ± 8%). (**b**) SsrB H12K only showed a moderate increase in activity at pH 5.6 (2.7-fold); however, the SsrB H12Y and H12F substitutions retained an acid-stimulated increase in the activity (6.9- and 6.7-fold, respectively) comparable to the wild-type (5-fold). (**c**) $P_{sifA}$-$lacZ$ activity measured from strains recovered at 8 hr post HeLa infection. The activity of the SsrB H12Q and SsrB H12Y-expressing strains was 62 ± 12% and 58 ± 3%, respectively, relative to the SsrB-expressing strain (p<0.001, n = 3). (**d**) Intracellular survival of strains in HeLa cells at 16 hr post infection (hpi). During HeLa cell infection, the c.f.u./ml of the SsrB-expressing strain increased 19-fold at 16 hpi. For SsrB H12Q and SsrB H12Y-expressing strains, the c.f.u./ml only increased 6.4- and 7.6-fold, respectively (p<0.0001, n = 3).

The online version of this article includes the following source data and figure supplement(s) for figure 3:

**Source data 1.** $P_{sifA}$-$lacZ$ activities and fold replication values for individual experiments used to create plots for **Figure 3**.

**Figure supplement 1.** DW637 strains expressing SsrB, SsrBc, H12Q, or H12Y infecting HeLa cells at 16 hr post infection (hpi).

*Figure 3 continued on next page*

*Figure 3 continued*

**Figure supplement 2.** Intracellular survival of 14028s strains expressing SsrB wild-type or H12Q or lacking SsrB during infection of RAW 264.7 macrophages.

**Figure supplement 2—source data 1.** Fold replication values for individual experiments used to create plots for *Figure 3—figure supplement 2*.

**Figure supplement 3.** Immunoblotting of strains expressing various SsrB constructs grown in magnesium minimal medium (MGM) pH 7.4.

**Figure supplement 3—source data 1.** Original uncropped images of immunoblot used to generate *Figure 3—figure supplement 2*.

that the SsrB protein in acid pH was highly oligomeric (*Liew et al., 2019*). As the promoter recognition site of SsrB is a degenerate A-T-rich 18 bp palindrome (*Walthers et al., 2007*; *Tomljenovic-Berube et al., 2010*), this increased affinity towards a SPI-2 promoter at acid pH is designed to selectively activate SPI-2 transcription within the macrophage vacuole during infection.

The non-conserved histidines His28, His34, and His72 are located in solvent-exposed regions in the predicted structure of SsrB. Their substitution did not have any effect on the acid-stimulated DNA binding of SsrB. Surprisingly, it was the relatively conserved residue His12, which is solvent-accessible and in the vicinity of active site residues that confers pH sensing to SsrB. In the i-TASSER and Alpha-Fold predicted structures of SsrB, His12 can form an H-bond with Asp11; it could also form a π–cation interaction with Lys106 (*Figure 5—figure supplement 1*; *Jumper et al., 2021*; *Yang et al., 2015*). In a glutamate substitution (H12Q), the H-bond with Asp11 could potentially be maintained, and this could contribute to the full activity we observed at neutral pH (*Figure 3b*). In the tyrosine/phenylal-anine substitutions, the π–cation interaction with Lys106 would be maintained and could potentially contribute to the acid pH-induced increase in SsrB activity (*Nakajima et al., 2011*). While the aromatic nature of the imidazole ring appears to contribute to pH sensing in SsrB, a role for His12 protonation at acid pH cannot be ruled out as the basic amino acid substitutions lysine and arginine could possess reduced activity due to steric hindrance, making it difficult to assess the effect of protonation changes on activity. It appears that the mechanism behind SsrB pH sensing involves the aromatic nature of the imidazole side chain of His12, and this side chain may interact with Lys106 in the active site to influence activity. Structural analysis will hopefully reveal the interactions of His12 with other residues in the context of pH sensing.

As His12 is located near the SsrB active site, it was unsurprising that it influenced SsrB phosphoryla-tion. The His12Q substitution of SsrB required higher PA concentrations for maximal phosphorylation at neutral pH compared to the wild-type. At higher PA and magnesium concentrations, H12Q phos-phorylation remained unchanged between neutral and acid pH, and it was similar to the wild-type at acid pH. While His12 substitutions have been reported to reduce autophosphorylation rates in RcsB and DegU (*Casino et al., 2018*; *Dahl et al., 1992*), we demonstrate its influence on pH dependence of RR phosphorylation for the first time. This result is in contrast to RcsB, where it was reported that a His12A substitution reduced phosphorylation by 90% (*Casino et al., 2018*). In the crystal structures of the NarL/FixJ subfamily RRs RcsB, VraR, and LiaR solved in the presence of the phosphomimetic $BeF_3$, the conserved histidine residue is oriented in such a way that it can participate in a π–cation interaction with the $Mg^{2+}$ ion essential for phosphorylation (*Casino et al., 2018*; *Davlieva et al., 2016*; *Kumar et al., 2022*; *Leonard et al., 2013*). In the unphosphorylated form, the histidine side chain of these same RRs faces away from the active site. Changes in the His12 rotameric state could thus modulate phosphorylation via its ability to coordinate the $Mg^{2+}$ ion. Structural experiments will hopefully shed light on the interaction between $Mg^{2+}$ and His12.

The DNA-binding activity of full-length SsrB is highly cooperative at the SPI-2 promoter *sseI*. Although SsrBc can form a dimer (*Carroll et al., 2009*), its Hill coefficient was substantially lower than the wild-type (*Figure 1a–d*). This result highlights a heretofore unappreciated property of the SsrB receiver domain in the formation of higher order structures that are important for a robust level of transcription observed within the vacuole. Substitution of His12 also reduced cooperative binding at acid pH. In an AlphaFold structure of the SsrB dimer, residues Asp11, His12, and Lys106 are all present at the dimer interface (*Figure 5—figure supplement 2a and b*). The presence of His12 at the dimer interface has also been observed in crystal structures of RcsB, VraR, and LiaR (*Casino et al., 2018*; *Huesa et al., 2021*; *Kumar et al., 2022*; *Leonard et al., 2013*). In the structure of an SsrB dimer bound to DNA, modeled using a similar RcsB crystal structure (PDB: 6ZIX), the dimer interface involves both the receiver domain and the C-terminal DNA-binding domain (*Figure 5—figure supplement*

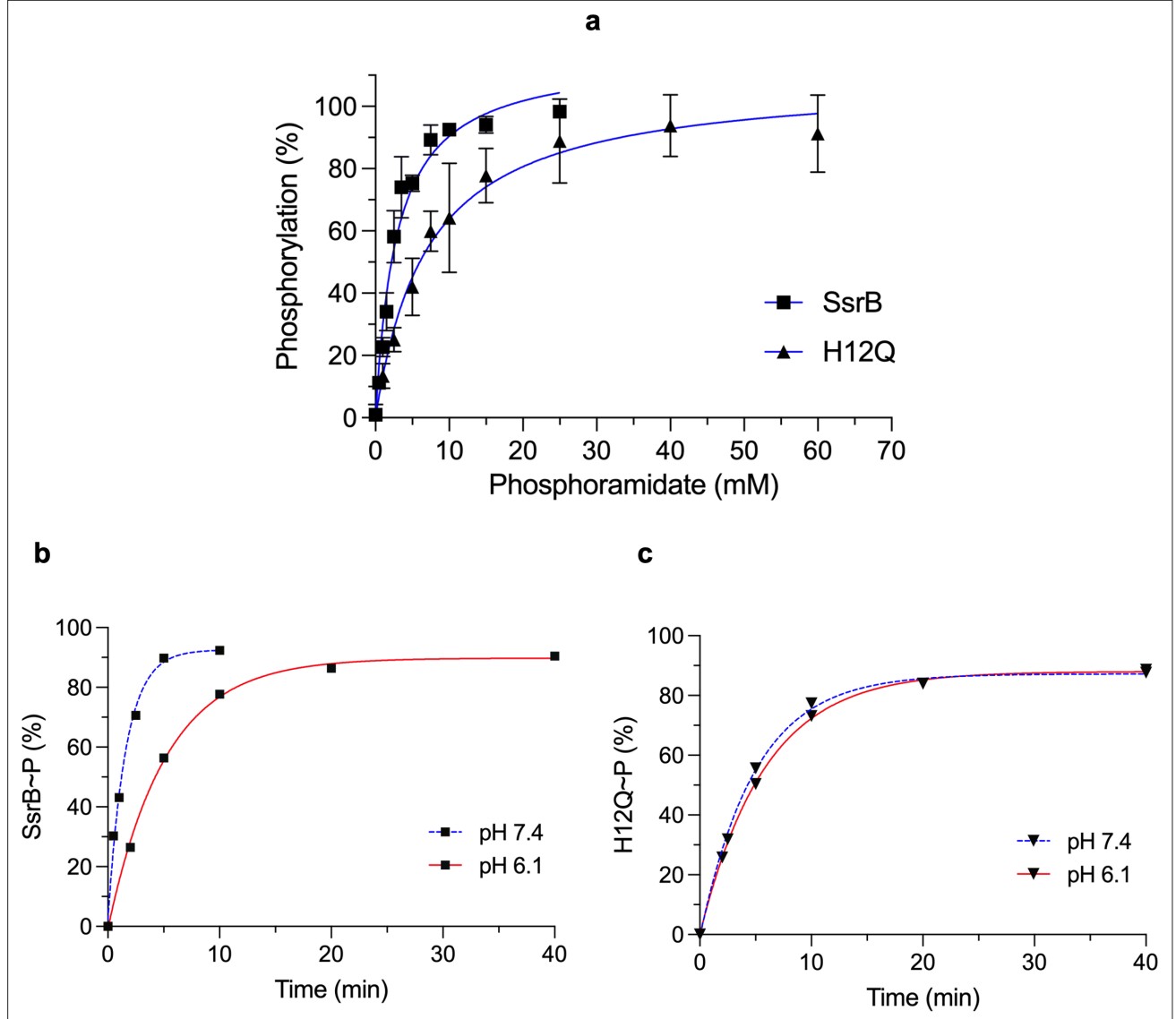

**Figure 4.** His12 substitution reduces SsrB phosphorylation. (**a**) The plot represents the percentage of phosphorylated protein (12 µM) at pH 7.4 after 10 min as a function of phosphoramidate (PA) concentration (mM) in the phosphorylation reaction. The $K_{0.5}$ for SsrB (squares) and H12Q (triangles) phosphorylation with PA was 2.9 ± 0.3 and 7.1 ± 1.1 mM, respectively. The error bars represent the standard deviation of two independent measurements. The blue lines represent the curve derived from fitting the points to the hyperbolic equation to determine the $K_{0.5}$. The absence of error bars indicates that the standard deviation was < the symbol. (**b**) The plots represent the percentage of phosphorylated protein (12 µM) with 15 mM PA and 100 mM $MgCl_2$ at varying times at neutral (blue) or acid pH (red) (see 'Materials and methods'). Phosphorylation of SsrB slows down at acid pH ($K_{obs}$ = 0.19 min⁻¹) compared to neutral pH ($K_{obs}$ = 0.62 min⁻¹). (**c**) The rate of H12Q phosphorylation was similar at neutral and acid pH ($K_{obs}$ = 0.20 and 0.17 min⁻¹, respectively). The blue or red lines represent the curve derived from fitting the points to a single exponential decay equation to determine the $K_{obs}$ (n=1).

The online version of this article includes the following source data and figure supplement(s) for figure 4:

**Source data 1.** Phosphorylated protein percentages for individual experiments used to create plots for **Figure 4**.

**Figure supplement 1.** The elution profile of unphosphorylated and phosphorylated SsrB and SsrB H12Q in a 40–50% acetonitrile in water gradient.

---

*2c*). Both the models align well at the receiver domain and highlight the importance of His12 in the formation of the SsrB dimer. Based on our binding data (*Figure 2d and e*) and the structural models, His12 is required for SsrB oligomerization at acidic pH. A crystal structure of unphosphorylated RcsB (PDB id: 5O8Y) showed an asymmetric unit containing six molecules of RcsB forming three dimers arranged in a hexameric assembly resembling a cylinder (*Casino et al., 2018*). Interestingly, the receiver domain from one subunit dimerized with the DNA-binding domain from another subunit in a

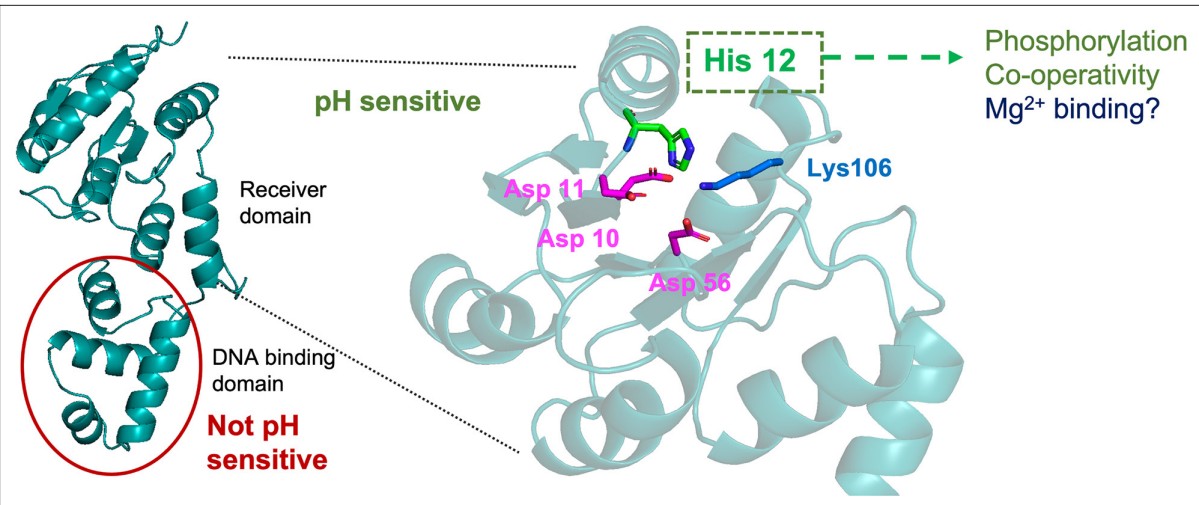

**Figure 5.** Summary of key findings. SsrB pH sensing lies in the receiver domain and not in the DNA-binding domain. A conserved histidine at position 12, which lies in the proximity of active site residues, is responsible for pH sensing. His12 also affects phosphorylation and cooperativity of DNA binding, and potentially influences $Mg^{2+}$ binding as well.

The online version of this article includes the following figure supplement(s) for figure 5:

**Figure supplement 1.** His12 interactions with residues in the SsrB receiver domain.

**Figure supplement 2.** AlphaFold prediction of an SsrB dimer.

**Figure supplement 3.** A model for an SsrB oligomer, generated using the RcsB hexamer (PDB id: 5O8Y) as the template.

crossed conformation. In this crossed conformation, His12 interacted with Glu170 in the DNA-binding domain, suggesting a scenario as to how His12 in the N-terminus can allosterically modify the affinity for DNA of the C-terminus. We modeled an SsrB oligomer using the RcsB hexamer as the template (*Figure 5—figure supplement 3*). Glu172 may potentially interact with His12 to contribute to the hexamer interface. Additional residues in the receiver and DNA-binding domain that participate in interchain contacts in the hexamer would be interesting to study in terms of their influence on SsrB cooperativity. It is worth noting that the RcsB His12A substitution behaved substantially differently from SsrB His12Q as RcsB His12A was not phosphorylated and was unable to bind DNA (*Casino et al., 2018*), whereas His12Q of SsrB binds DNA with an affinity comparable to that of the wild-type at neutral pH (*Figure 2d*). These findings again highlight the important functional differences of RR homologues.

This work thus identifies the conserved His12 as playing a critical role in pH sensing, and formation of higher order structures in SsrB and highlights its importance during *Salmonella* infection (*Figure 5*). Whether or not this conserved histidine plays a similar role in other NarL/FixJ subfamily RRs has been largely unexplored to date but would seem likely based on its high degree of conservation and the existing structural data. It was surprising that pH sensing appeared to be solely determined by a single amino acid in the phosphorylation domain of SsrB. Eliminating pH sensing via His12 substitution rendered *Salmonella* completely avirulent and unable to survive and replicate in the vacuole (*Figure 3*). Preventing pH switching as a means of controlling virulence is thus a novel strategy for controlling *Salmonella* pathogenesis.

## Materials and methods
### Strains and growth conditions

Bacterial strains and plasmids used in this study are listed in *Table 2*. ST strains were grown in either Lysogeny Broth (Difco) or MGM: 100 mM Tris-HCl or MES-NaOH for pH 7.4 and 5.6, respectively, 5 mM KCl, 7.5 mM $(NH_4)_2SO_4$, 0.5 mM $K_2SO_4$, 1 mM $KH_2PO_4$, 8 µM $MgCl_2$, 0.2% (w/v) glucose, and 0.1% (w/v) casamino acids (*Beuzón et al., 1999*). The bacteria were grown at 37°C with shaking at 250 rpm. Transformation of plasmids carrying SsrBc or SsrB constructs into ST was performed using electroporation. HeLa cells and RAW macrophages were maintained in Dulbecco's modified Eagle

**Table 2.** Cell lines, strains, and plasmids used in this study.

| Strain/plasmid/cell line | Description/genotype | Reference/ source |
|---|---|---|
| *Salmonella strains* | | |
| 14028s WT | *Salmonella enterica* serovar Typhimurium strain 14028s | Lab stock |
| 14028s.*ssrB.H12Q* | 14028s *ssrB:ssrB.H12Q-catR* | This work |
| DW637 | 14028s Δ*ssrB attP::P*sifA*-lacZ* (Kan^R) | *Desai et al., 2016* |
| *E. coli strains* | | |
| TOP10 | *E. coli* DH10B | Saunders lab, MBI |
| | | |
| *Plasmids* | | |
| pMPMA5Ω | (Amp^R) | Lab stock |
| pMPM_SsrBWT | pMPMA5Ω plasmid cloned with 6xhis-*ssrB* (Amp^R) | *Feng et al., 2004* |
| pMPM_SsrBc | pMPMA5Ω plasmid cloned with 6xhis-*ssrBc* (Amp^R) | *Feng et al., 2004* |
| pMPM_SsrB_H28A | pMPMA5Ω plasmid cloned with 6xhis-*ssrB.H28A* (Amp^R) | This work |
| pMPM_SsrB_H34A | pMPMA5Ω plasmid cloned with 6xhis-*ssrB.H34A* (Amp^R) | This work |
| pMPM_SsrB_H72A | pMPMA5Ω plasmid cloned with 6xhis-*ssrB.H72A* (Amp^R) | This work |
| pMPM_SsrB_2H1 | pMPMA5Ω plasmid cloned with 6xhis-*ssrB.H28A.H34A* (Amp^R) | This work |
| pMPM_SsrB_2H2 | pMPMA5Ω plasmid cloned with 6xhis-*ssrB.H28A.H72A* (Amp^R) | This work |
| pMPM_SsrB_2H3 | pMPMA5Ω plasmid cloned with 6xhis-*ssrB.H34A.H72A* (Amp^R) | This work |
| pMPM_SsrB_3H | pMPMA5Ω plasmid cloned with 6xhis-*ssrB.H28A.H34A.H72A* (Amp^R) | This work |
| pMPM_SsrB_H12A | pMPMA5Ω plasmid cloned with 6xhis-*ssrB.H12A*(Amp^R) | This work |
| pMPM_SsrB_H12Q | pMPMA5Ω plasmid cloned with 6xhis-*ssrB.H12Q* (Amp^R) | This work |
| pMPM_SsrB_H12N | pMPMA5Ω plasmid cloned with 6xhis-*ssrB.H12N* (Amp^R) | This work |
| pMPM_SsrB_H12Y | pMPMA5Ω plasmid cloned with 6xhis-*ssrB.H12Y* (Amp^R) | This work |
| pMPM_SsrB_H12K | pMPMA5Ω plasmid cloned with 6xhis-*ssrB.H12K* (Amp^R) | This work |
| pMPM_SsrB_H12F | pMPMA5Ω plasmid cloned with 6xhis-*ssrB.H12F* (Amp^R) | This work |
| pKDS121 | pKD3 plasmid with *ssrB.H12Q* inserted upstream the *catR* gene | This work |
| *Cell lines* | | |
| HeLa | Human Cervical Adenocarcinoma cell line | ATCC |
| RAW264.7 | Murine Macrophage cell line | Bruno lab, UTMB |

medium (DMEM) containing sodium pyruvate, essential amino acids, 10% (v/v) fetal bovine serum (Gibco), and 1% (v/v) Penicillin/Streptomycin (P/S) at 37°C and 5% $CO_2$.

## Molecular biology techniques, cloning, and protein purification

DNA manipulations and cloning were performed using standard protocols described in *Sambrook and Russell, 2001*. Enzymes, reagents, and DNA purification kits were purchased from Thermo Fisher, New England Biolabs, and QIAGEN. Oligonucleotides were ordered from IDT Asia and IDT US and are listed in *Table 3*. SsrB histidine substitutions were generated by performing site-directed PCR mutagenesis using the pMPMA5Ω-SsrB wild-type plasmid (*Feng et al., 2004*) as the template. The 14028s.*ssrB.H12Q* strain was generated using the lambda red recombination technique (*Karlinsey, 2007*). Briefly, the *ssrB.H12Q-catR* fragment was generated by inserting the *ssrB.H12Q* gene into the pKD3 plasmid upstream of the *catR* gene using the NEBuilder HiFi DNA Assembly Kit. This fragment was then amplified with 40 bp of homology region upstream and downstream of the *ssrB* coding

**Table 3.** Primers and oligonucleotides used in this study.

| Primer/oligo | Description | Sequence |
|---|---|---|
| SsrB_His28A_FP | SsrB: substituting His28 to A | TTACCCTGGCCTGCCTTTAAAATTGTA |
| SsrB_His28A_RP | SsrB: substituting His28 to A | TACAATTTTAAAGGCAGGCCAGGGTAA |
| SsrB_His34A_FP | SsrB: substituting His34 to A | TTTAAAATTGTAGAGGCGGTTAAAAATGGTCTT |
| SsrB_His34A_RP | SsrB: substituting His34 to A | AAGACCATTTTTAACCGCCTCTACAATTTTAAA |
| SsrB_His72A_FP | SsrB: substituting His72 to A | ATTCCTCAATTAGCACAGCGTTGGCC |
| SsrB_His72A_RP | SsrB: substituting His72 to A | GGCCAACGCTGTGCTAATTGAGGAAT |
| SsrB_H12N_FP | SsrB: substituting His12 to N | TTAGTAGACGATAATGAAATCATCA |
| SsrB_H12N_RP | SsrB: substituting His12 to N | GATGATTTCATTATCGTCTACTAATAA |
| SsrB_H12K_FP | SsrB: substituting His12 to K | TTAGTAGACGATAAGGAAATCATCA |
| SsrB_H12K_RP | SsrB: substituting His12 to K | GATGATTTCCTTATCGTCTACTAATAA |
| SsrB_H12Y_FP | SsrB: substituting His12 to Y | TTAGTAGACGATTATGAAATCATCA |
| SsrB_H12A_FP | SsrB: substituting His12 to A | TTAGTAGACGATGCGGAAATCATCA |
| SsrB_H12Q_FP | SsrB: substituting His12 to Q | TTAGTAGACGATCAGGAAATCATCA |
| SsrB_H12F_FP | SsrB: substituting His12 to F | TTAGTAGACGATTTTGAAATCATCATT |
| SsrB_RP | SsrB reverse primer | TTAATACTCTATTAACCTCATTCTTCG |
| pKDSsrB_FP | To amplify *ssrB.H12Q* with homology to pKD3 | ATATGAATATCCTCCTATGAAAGAATATAAGATCTTATTA |
| H12Q_CmR_RP1 | To amplify *ssrB.H12Q* with homology upstream of *catR* | GGACCATGGCTAATTCCCATTTAATACTCTATTAA |
| GA_CmR_FP | To amplify pKD3 with homology to *ssrB.H12Q* | TTAATAGAGTATTAAATGGGAATTAGCCATGGTCC |
| pKD3_FP | To amplify pKD3 with homology to *ssrB.H12Q* | GATCTTATATTCTTTCATACTAAGGAGGATATTCATAT |
| Lam_FP | To amplify *ssrB.H12Q.catR* with 40 bp homology upstream of *ssrB* in 14028s | ATTACTTAATATTATCTTAATTTTCGCGAGGGCAGCAAAATGAAAGAATA |
| Lam_RP2 | To amplify *ssrB.H12Q.catR* with 40 bp homology downstream of *ssrB* in 14028s | CAAAATATGACCAATGCTTAATACCATCGGACGCCCCTGGGTGTAGGCTGGAG |
| Ex_SsrB_F | To confirm insertion of *ssrB.H12Q.catR* in 14028s | TTGGTATGCTATGTCATAGACA |
| Ex_SsrB_R | To confirm insertion of *ssrB.H12Q.catR* in 14028s | GTGCGGCATACCAGGGCATC |
| pMPM_FP | For sequence check of constructs in pMPMA5Ω | TACCTGACGCTTTTTATCGC |
| pMPM_RP | For sequence check of constructs in pMPMA5Ω | CTTCTCTCATCCGCCAAAAC |
| *sseI* hairpin | Used for single-molecule unzipping assay | 5'-Biotin-TTTTTTTTTTCTATGCGCCAGTCCTTAATGGCATTATCTGAATCGTTAAGTAATTTCTTGTG TGAAATTA CTTAACGATTCAGATAATGCCATTAAGGACTGGCGCATAGAGCAGCGTCCCGGGCGGCC |

region and the purified PCR product was transformed into electrocompetent 14028s strains carrying the pKD46 plasmid. Recombinant colonies were selected on a chloramphenicol plate, the insertion was confirmed by PCR, and selected colonies were subsequently grown at 42°C to expel the pKD46 plasmid. Sequences of the final constructs were verified by Axil Scientific, Singapore or Genewiz, USA. Cloning and maintenance of plasmids used *Escherichia coli* DH10B (TOP10). His$_6$-tagged SsrB, SsrB H12Q, and SsrBc were purified as described previously (*Carroll et al., 2009*).

## Single-molecule unzipping assay of DNA binding

The DNA hairpin was prepared as described previously (*Gulvady et al., 2018*) using the 47 bp region of the *sseI* promoter that was protected from DNase I in the presence of SsrB (*Feng et al., 2004*;

sequence in *Table 3*). Preparation of the flow channel, attaching the DNA hairpin to the channel and the single-molecule unzipping assay, was performed as described previously (*Gulvady et al., 2018*; *Liew et al., 2019*). Briefly, in each experiment, the minimum hairpin unzipping force, known as the critical force, was determined in the absence of the protein. Binding events, represented by a delay in unzipping of the hairpin at the critical force, in the presence of a given concentration of SsrB were measured over the course of 32 force cycles at pH 7.4 (50 mM KCl, 10 mM Tris) or 6.1 (50 mM KCl, 10 mM MES). The assay was then repeated with purified SsrB H12Q or SsrBc at pH 7.4 or 6.1. The probability of binding events was plotted as a function of the protein concentration, and the curve obtained by fitting the graph to the Hill equation was used to determine the $K_D$ and Hill coefficients ($n$). Three to five independent hairpins were assayed for each protein concentration. For the binding curve of SsrB at pH 7.4, we plotted two more curves using the upper and lower limits of the binding probabilities for the 150 nM concentration and determined their $K_D$ and $n$ values (*Figure 1—figure supplement 1*).

## $\beta$-Galactosidase assays

$\beta$-Galactosidase assays were performed as described previously (*Desai et al., 2016*; *Feng et al., 2003*). DW637 strains carrying the pMPMA5Ω-SsrBc or SsrB constructs were grown overnight in LB containing 100 µg.ml$^{-1}$ Ampicillin, sub-cultured 1:100 in 5 ml MGM pH 7.4 or pH 5.6. At early exponential phase ($OD_{600nm}$ = 0.36–0.38, 3 hr 10 min), arabinose was added to a final concentration of 0.1% to the cultures. At 3 hr post induction, the $OD_{600}$ of 200 µl of the diluted culture (1:2) was measured at 595 nm in a 96-well microtiter plate. Then, 50 µl of this diluted culture was added to 450 µl lysis buffer containing 0.01% SDS, 50 mM $\beta$-mercaptoethanol, and 30 µl CHCl$_3$ in Z-buffer (*Miller, 1972*). The mixture was vortexed and incubated on a nutator for 5 min to lyse the cells. Also, 100 µl of ONPG (ortho-Nitrophenyl-$\beta$-galactoside, 4 mg.ml$^{-1}$ in Z-buffer) was added to this mixture and incubated at 37° C until a yellow-colored product was formed. The reaction was stopped by adding 250 µl of 1 M Na$_2$CO$_3$. Then, 200 µl of the reaction mixture supernatant was transferred to a 96-well microtiter plate, and the OD was measured at 415 nm and 550 nm. The $\beta$-galactosidase activity was determined using the formula

$$\text{Activity(Miller units)} = \frac{1000 \times (OD_{415nm} - 1.75 \times OD_{550nm})}{OD_{415nm} \times \text{time of reaction(min)} \times \text{volume of culture(ml)}}$$

A minimum of three independent experiments were performed for each SsrB construct.

## Immunoblotting

DW637 strains carrying pMPMA5Ω, SsrB WT, H12Q, or H12Y were grown in MGM pH 7.4 (10 ml each) and induced for the $\beta$-galactosidase assay as described above. Bacteria were harvested by centrifugation at 16,000 × $g$, washed once and resuspended in Tris buffer (pH 6.8), and sonicated (Fisher Scientific Sonic Dismembrator Model 100, setting 4, three pulses of 15 s). The lysed suspension was centrifuged at 16,000 × $g$, 4°C for 20 min to separate the debris, and the supernatant was used as the cell-free extract. The cell-free extract was mixed with 4× SDS-PAGE dye in a 3:1 ratio and heated at 95°C for 10 min. The samples were then separated on a gradient gel (4–20% Mini-PROTEAN TGX Precast Protein Gels) with standard running buffer (0.4% [w/v] SDS, 25 mM Tris, and 192 mM glycine) at 125 V. The separated samples were transferred to a PVDF membrane (Trans-Blot Turbo Mini 0.2 µm PVDF Transfer Packs) through semi-dry transfer (Bio-Rad Trans-Blot Turbo Transfer System). The PVDF membrane was blocked for 1 hr at room temperature with 5% (w/v) bovine serum albumin (BSA) in Tris-buffered saline containing 0.1% (w/v) Triton-X-100 (TBS-T). The membrane was cut into two parts between the 35 and 55 kDa molecular weight markers. The lower molecular weight part of the membrane was incubated overnight at 4°C in anti-6X-HisTag antibody (1:1000 dilution in blocking buffer, Invitrogen MA1-21315-HRP) to stain for SsrB protein constructs. The higher molecular weight part of the membrane was incubated overnight at 4°C in anti-DNAK antibody (1:5000 dilution in blocking buffer, Abcam #69617 mouse monoclonal [8E2/2]) as the loading control. Both membranes were given five washes with TBS-T at room temperature. The higher molecular weight membrane was incubated for 1 hr with HRP-tagged anti-mouse IgG (1:10,000 dilution in 1% [w/v] BSA in TBS-T, Santa Cruz Biotech goat anti-mouse IgG-HRP) at room temperature, followed by three washes with

TBS-T. Both membranes were visualized using Clarity Max Western ECL Substrate (*Figure 3—figure supplement 3*).

## HeLa cell survival and $\beta$-galactosidase assays

HeLa cell infections were performed as described previously (*Walthers et al., 2007*). For the intracellular survival assay, $5 \times 10^4$ HeLa cells were seeded and grown for 24 hr in 24-well plates, washed twice with Dulbecco's phosphate-buffered saline (DPBS), and incubated in 0.5 ml DMEM without P/S containing 0.1% arabinose before infection. Overnight grown cultures of DW637 strains carrying pMPMA5Ω-SsrBc or SsrB constructs were diluted 1:30 in 3 ml LB and grown until late stationary phase. The bacteria were diluted to a final $OD_{600}$ of 0.2 in DPBS. Then, 83 μl of the diluted bacterial culture was added to the HeLa cells (multiplicity of infection [MOI] = 200) and incubated for 30 min. To remove the unattached bacteria in the culture medium, the cells were washed twice with DPBS. Fresh DMEM containing 0.1% (w/v) arabinose and 100 μg.ml$^{-1}$ gentamicin was added (0 hpi) and the cells were incubated for 1 hr. For the rest of the infection, the cells were incubated in DMEM containing 0.1% (w/v) arabinose and 20 μg.ml$^{-1}$ gentamicin. Intracellular ST were harvested at 2 and 16 hpi as follows: HeLa cells were washed with DPBS and incubated in 0.5 ml of 0.1% (w/v) Triton-X-100 in PBS for 10 min. After mixing by vigorous pipetting, 100 μl of the lysate was used to prepare serial dilutions in PBS for viable counting.

For the intracellular $\beta$-galactosidase assay, $2 \times 10^5$ HeLa cells were seeded in 6-well plates and infection was performed as described above while increasing the volume proportionately. At 8 hpi, cells were washed once with DPBS and incubated in 1 ml of 0.1% (w/v) Triton-X-100 in PBS for 10 min. The mixture was vigorously mixed and centrifuged at $500 \times g$ for 30 s at 4°C. The supernatant was centrifuged at $16,000 \times g$ for 10 min and washed once with PBS. The final pellet was resuspended in 120 μl of PBS, of which 10 μl was used to prepare serial dilutions for viable counting and 100 μl was used to perform the $\beta$-galactosidase assay. The final activity was measured as follows: Activity = (OD of the reaction mixture at 420 nm × $10^9$) ÷ (Time of reaction in min × colony-forming units.ml$^{-1}$). A minimum of three independent experiments were performed for each SsrB construct.

## RAW macrophage infection

RAW macrophage infections were performed as follows. For the intracellular survival assay, $1 \times 10^5$ cells were seeded and grown for 24 hr in 24-well plates, washed twice with DPBS and incubated in 0.5 ml DMEM without P/S. Single colonies of 14028s wild-type, 14028s *ssrB.H12Q,* or 14028s *ΔssrB* were grown overnight in 5 ml LB. The bacteria were diluted to a final MOI = 5 in DMEM, and this media was added to the macrophages. The infection was synchronized by centrifugation at 600 × g for 5 min, followed by an incubation of 30 min. To remove the unattached bacteria in the culture medium, the macrophages were washed twice with DPBS and incubated in fresh DMEM containing 100 μg.ml$^{-1}$ gentamicin for 1 hr. For the remainder of the infection, the macrophages were incubated in DMEM containing 20 μg.ml$^{-1}$ gentamicin. Intracellular ST were harvested at 2 and 16 hpi.

## Phosphorylation assay

All phosphorylation assays were performed at room temperature using a phosphorylation buffer of pH 7.4 or 6.1, with PA as the phosphodonor (PA preparation: *Sheridan et al., 2007*). Purified SsrB WT or SsrB H12Q was added to a 100 μl reaction at a final concentration of 12 μM. For the time-dependent reactions, the reaction mixture contained 1× phosphorylation buffer (50 mM HEPES-NaOH pH 7.4 or 50 mM MES-NaOH pH 6.1, 100 mM MgCl$_2$, appropriate amounts of KCl for a final ionic strength of 430 mM) with 2.5 mM PA. The reaction was immediately mixed after adding PA and incubated for the specified amount of time. The reaction was stopped by addition of 10 μl of EDTA (final concentration 50 mM), centrifuged for 1 min at $20,000 \times g$, and 82.5 μl of the supernatant was injected into an HPLC instrument (Waters 1525 binary pump) with a $C_4$ reverse-phased column (Vydac 214TP54) and water:acetonitrile (containing 0.1% [v/v] trifluoroacetic acid) solvent system. The following method was used: 40% (v/v) acetonitrile (5 min), 40–50% (v/v) acetonitrile gradient (20 min), 90% (v/v) acetonitrile (7 min), 40% (v/v) acetonitrile (8 min). The area under each observed peak was analyzed with Breeze software. The plot of the percentage of phosphorylated protein against time was fitted to a single exponential decay to obtain the $B_{max}$ and $K_{obs}$ using Prism software. For the PA concentration-dependent reactions, the reaction mixture contained 12 μM SsrB WT or

SsrB H12Q, 1× phosphorylation buffer (50 mM HEPES-NaOH pH 7.4 or 50 mM MES-NaOH pH 6.1, 50 mM KCl, 20 mM MgCl$_2$), with the appropriate amount of PA (0.5–100 mM final concentration). The plot of the percentage of phosphorylated protein against the PA concentration was fitted to a hyperbolic curve using Prism to obtain the amount of PA required to phosphorylate 50% of the protein ($K_{0.5}$).

## Acknowledgements

We acknowledge helpful discussions regarding SsrB structure with Mark White (Sealy Center for Structural Biology, UTMB) and for *Figure 5—figure supplements 2 and 3*. We are grateful to Dr. Ranjit Gulvady for advice and instruction on the single-molecule experiments and Prof. Yan Jie (MBI, NUS, Singapore) for providing laboratory space. This work was supported in part by a Regional Centre of Excellence Grant from the Ministry of Education to the Mechanobiology Institute, National University of Singapore. LJK acknowledges myopic PCMB reviewers of this project. LJK and DS were also supported by start-up funds from UTMB, a Texas STAR award and CPRIT RP200650 to LJK.

## Additional information

### Funding

| Funder | Grant reference number | Author |
|---|---|---|
| Mechanobiology Institute, Singapore | Regional centre of excellence | Dasvit Shetty |
| University of Texas Medical Branch | | Linda J Kenney |
| Cancer Prevention and Research Institute of Texas | RP200650 | Linda J Kenney |
| Texas STAR | | Linda J Kenney |

The funders had no role in study design, data collection and interpretation, or the decision to submit the work for publication.

### Author contributions

Dasvit Shetty, Conceptualization, Data curation, Formal analysis, Validation, Investigation, Visualization, Methodology, Writing – original draft, Writing – review and editing; Linda J Kenney, Conceptualization, Resources, Supervision, Funding acquisition, Writing – original draft, Writing – review and editing

### Author ORCIDs
Dasvit Shetty ⓘ http://orcid.org/0000-0001-8778-3615
Linda J Kenney ⓘ http://orcid.org/0000-0002-8658-0717

### Decision letter and Author response
Decision letter https://doi.org/10.7554/eLife.85690.sa1
Author response https://doi.org/10.7554/eLife.85690.sa2

## Additional files

### Supplementary files
• MDAR checklist

### Data availability
All data generated or analysed during this study are included in the manuscript and supporting files; Source Data files have been provided for Figures 1-4, and figure supplements for Figures 2 and 3.

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
