## [Editor Report]

*Salmonella* invades and survives in host cells via SPI-1 and SPI-2 type 3 secretion system mechanisms, with the SPI-2 system allowing for intracellular survival in *Salmonella*-containing vacuoles, which have a low-pH environment. Transcription of SPI-2 genes at low pH is activated by the DNA-binding SsrB protein, which sits at the top of the SPI-2 regulatory hierarchy. This important study provides convincing insights as to how SsrB is allosterically affected by pH, resulting in acid-dependent DNA binding.

---

## [Decision Letter]

**Decision letter after peer review:**

Thank you for submitting your article "A pH-sensitive switch activates virulence in *Salmonella*" for consideration by *eLife*. Your article has been reviewed by 3 peer reviewers, and the evaluation has been overseen by a Reviewing Editor and Arturo Casadevall as the Senior Editor. The following individual involved in the review of your submission has agreed to reveal their identity: Kelly Hughes (Reviewer #1).

Essential revisions:

As you can see the reviews were mixed. Reviewer 2 thought that the conclusions were not fully supported by the data and raised a series of criticisms that need to be addressed with new experimental data. In addition, reviewers 1 and 3 have important suggestions for how to improve this paper. We will consider a revision, but it needs to address all the comments of the review and the paper will be sent out for re-review to the same reviewers.

*Reviewer #1 (Recommendations for the authors):*

Recommendation for the authors – none; solid, high-quality data that provides a fundamental molecular mechanistic basis for pH sensing and the low-pH activation of SPI-2 gene transcription by SsrB that is critical for *Salmonella* survival within host cells.

*Reviewer #2 (Recommendations for the authors):*

1. Line 113 and beyond – A measurement is only as accurate as the least accurate part of the experiment. Throughout the manuscript, many results are given with three or four significant figures, implying accuracy to within one part in 1,000 or 10,000. I would argue that experiments involving the measurement of liquid volumes are at best accurate to one part in 100 and therefore data should be reported using no more than two significant figures.

*Reviewer #3 (Recommendations for the authors):*

One limitation of the study is that the functional consequences of SsrB His12 mutants (unable to sense pH changes and with reduced SsrB phosphorylation) were only measured in HeLa cells. Although this cell line is a valid choice for proof-of-principle studies like this one, the use of other tissue culture models (intestinal epithelial cells, bone marrow-derived macrophages) would have been more relevant for *Salmonella* pathogenesis. Moreover, the use of an animal model of infection (e.g., intraperitoneal infection experiments comparing *Salmonella* WT, an SsrB deletion mutant, and one or more SsrB His12 mutants) would further increase the impact of this study, by showing that a single His12 mutation is essential for *Salmonella* pathogenesis.

To further strengthen the conclusions, it would be nice to have data with more relevant cell types for *Salmonella* pathogenesis. in vivo experiments would also increase the impact of the manuscript. At the very minimum, some of the conclusions related to virulence (i.e. last paragraph of the discussion) need to be toned down and it should be discussed how future studies are necessary to elucidate the role of His12 and the SsrB pH switch for *Salmonella* pathogenesis.

[Editors' note: further revisions were suggested prior to acceptance, as described below.]

Thank you for resubmitting your work entitled "A pH-sensitive switch activates virulence in *Salmonella*" for further consideration by *eLife*. Your revised article has been evaluated by Detlef Weigel (Senior Editor) and a Reviewing Editor.

The manuscript has been improved but there are some remaining issues that need to be addressed, as outlined in the comments of reviewer #2 below. Thank you!

*Reviewer #2 (Recommendations for the authors):*

I was Reviewer #2 for the original version of this manuscript. The authors are to be commended for thoroughly addressing my extensive concerns. The revised manuscript reports a comprehensive investigation of the role of His12 in pH sensing by the *Salmonella* response regulator SsrB. This is an interesting and important topic, and the evidence for all conclusions is now compelling.

Significant Figures

The authors did not address one concern raised previously, the use of significant figures in reporting results. This disagreement does not affect any conclusions in the manuscript but does imply to the reader a greater degree of accuracy than is warranted by the data, so I am going to point it out again. My original comment was:

"A measurement is only as accurate as the least accurate part of the experiment. Throughout the manuscript, many results are given with three or four significant figures, implying accuracy to within one part in 1,000 or 10,000. I would argue that experiments involving measurement of liquid volumes are at best accurate to one part in 100 and therefore data should be reported using no more than two significant figures."

For example, reporting an inferred KD of 147.8 nM (four significant figures) in Figure 1a implies that the technique used could detect differences in KD of 0.1 nM, whereas I would argue the value should be reported as a KD of 150 nM (two significant figures), implying that differences of 10 nM are detectable. The KD value depends on the accuracy of the protein concentration used, which I believe is unlikely to be accurate to better than 1%.

---

## [Author Response]

Essential revisions:Reviewer #2 (Recommendations for the authors):1. Line 113 and beyond – A measurement is only as accurate as the least accurate part of the experiment. Throughout the manuscript, many results are given with three or four significant figures, implying accuracy to within one part in 1,000 or 10,000. I would argue that experiments involving the measurement of liquid volumes are at best accurate to one part in 100 and therefore data should be reported using no more than two significant figures.

Corrected.

Reviewer #3 (Recommendations for the authors):One limitation of the study is that the functional consequences of SsrB His12 mutants (unable to sense pH changes and with reduced SsrB phosphorylation) were only measured in HeLa cells. Although this cell line is a valid choice for proof-of-principle studies like this one, the use of other tissue culture models (intestinal epithelial cells, bone marrow-derived macrophages) would have been more relevant for *Salmonella* pathogenesis. Moreover, the use of an animal model of infection (e.g., intraperitoneal infection experiments comparing *Salmonella* WT, an SsrB deletion mutant, and one or more SsrB His12 mutants) would further increase the impact of this study, by showing that a single His12 mutation is essential for *Salmonella* pathogenesis.To further strengthen the conclusions, it would be nice to have data with more relevant cell types for *Salmonella* pathogenesis. in vivo experiments would also increase the impact of the manuscript. At the very minimum, some of the conclusions related to virulence (i.e. last paragraph of the discussion) need to be toned down and it should be discussed how future studies are necessary to elucidate the role of His12 and the SsrB pH switch for *Salmonella* pathogenesis.

The revised manuscript now includes an evaluation of His12 of SsrB in *Salmonella* infections of macrophages (new Figure 3—figure supplement 2).

[Editors' note: further revisions were suggested prior to acceptance, as described below.]

The manuscript has been improved but there are some remaining issues that need to be addressed, as outlined in the comments of reviewer #2 below. Thank you!Reviewer #2 (Recommendations for the authors):I was Reviewer #2 for the original version of this manuscript. The authors are to be commended for thoroughly addressing my extensive concerns. The revised manuscript reports a comprehensive investigation of the role of His12 in pH sensing by the *Salmonella* response regulator SsrB. This is an interesting and important topic, and the evidence for all conclusions is now compelling.Significant FiguresThe authors did not address one concern raised previously, the use of significant figures in reporting results. This disagreement does not affect any conclusions in the manuscript but does imply to the reader a greater degree of accuracy than is warranted by the data, so I am going to point it out again. My original comment was:"A measurement is only as accurate as the least accurate part of the experiment. Throughout the manuscript, many results are given with three or four significant figures, implying accuracy to within one part in 1,000 or 10,000. I would argue that experiments involving measurement of liquid volumes are at best accurate to one part in 100 and therefore data should be reported using no more than two significant figures."For example, reporting an inferred KD of 147.8 nM (four significant figures) in Figure 1a implies that the technique used could detect differences in KD of 0.1 nM, whereas I would argue the value should be reported as a KD of 150 nM (two significant figures), implying that differences of 10 nM are detectable. The KD value depends on the accuracy of the protein concentration used, which I believe is unlikely to be accurate to better than 1%.

Based on our data in Figure 1b, we can detect differences in binding at a minimum of 1.5 nM SsrB (compare data points 3.5 nM and 5 nM). We have updated Figures 1a, 1c, 1d, 2d, 2e, and Figure 1 supplement 1, with the exception of Figure 1b. Rounding off the *K_D_* value from 4.6 nM to 5 nM in Figure 1b will appear incorrect based on the figure, as the average binding probability at 5 nM is about ~60%, and the apparent K_D_ from the figure has a value between 4 and 5 nM.